# Modeling Impact Mechanics of 3D Helicoidally Architected Polymer Composites Enabled by Additive Manufacturing for Lightweight Silicon Photovoltaics Technology

**DOI:** 10.3390/polym14061228

**Published:** 2022-03-18

**Authors:** Arief Suriadi Budiman, Rahul Sahay, Komal Agarwal, Rayya Fajarna, Fergyanto E. Gunawan, Avinash Baji, Nagarajan Raghavan

**Affiliations:** 1Oregon Renewable Energy Center (OREC), Klamath Falls, OR 97601, USA; 2Department of Manufacturing and Mechanical Engineering and Technology, Oregon Institute of Technology, Klamath Falls, OR 97601, USA; 3Xtreme Materials Lab, Engineering Product Development, Singapore University of Technology and Design (SUTD), Singapore 487372, Singapore; komal_agarwal@alumni.sutd.edu.sg; 4Industrial Engineering Department, BINUS Graduate Program—Master of Industrial Engineering, Bina Nusantara University, Jakarta 11480, Indonesia; rayya.fajarna@binus.ac.id (R.F.); fgunawan@binus.edu (F.E.G.); 5Nano-Macro Reliability Lab, Engineering Product Development Pillar, Singapore University of Technology and Design (SUTD), Singapore 487372, Singapore; nagarajan@sutd.edu.sg; 6Department of Engineering, La Trobe University, Melbourne, VIC 3086, Australia; a.baji@latrobe.edu.au

**Keywords:** 3D helicoidal architecture, fiber-based polymer composite, impact resistance, lightweight photovoltaics (PV), numerical modeling

## Abstract

When silicon solar cells are used in the novel lightweight photovoltaic (PV) modules using a sandwich design with polycarbonate sheets on both the front and back sides of the cells, they are much more prone to impact loading, which may be prevalent in four-season countries during wintertime. Yet, the lightweight PV modules have recently become an increasingly important development, especially for certain segments of the renewable energy markets all over the world—such as exhibition halls, factories, supermarkets, farms, etc.—including in countries with harsh hailstorms during winter. Even in the standard PV module design using glass as the front sheet, the silicon cells inside remain fragile and may be prone to impact loading. This impact loading has been widely known to lead to cracks in the silicon solar cells that over an extended period of time may significantly degrade performance (output power). In our group’s previous work, a 3D helicoidally architected fiber-based polymer composite (enabled by an electrospinning-based additive manufacturing methodology) was found to exhibit excellent impact resistance—absorbing much of the energy from the impact load—such that the silicon solar cells encapsulated on both sides by this material breaks only at significantly higher impact load/energy, compared to when a standard, commercial PV encapsulant material was used. In the present study, we aim to use numerical simulation and modeling to enhance our understanding of the stress distribution and evolution during impact loading on such helicoidally arranged fiber-based composite materials, and thus the damage evolution and mechanisms. This could further aid the implementation of the lightweight PV technology for the unique market needs, especially in countries with extreme winter seasons.

## 1. Introduction

Lightweight photovoltaic (PV) modules are becoming increasingly important to certain sectors of the renewable energy market. Lightweight PV technology could potentially help address global climate and sustainability issues by being deployed in countries where electricity infrastructure is often lacking in very remote, poor locations separated by oceans [1,2,3,4]. For such a region, centralized energy sources may not be the best solution. By reducing the cost of transporting and installing solar PV systems, lightweight PV could play a critical part in building the kind of self-sufficient power infrastructure that is desperately desired in distant, rustic areas of developing countries. In order to be transported to remote locations with limited road and mobility infrastructure, PV power infrastructure must be lightweight. Lightweight PV modules are desired both for use in urban structures in advanced countries [5,6,7,8] and for easy set up in distant and impoverished locations in unusual areas of the developing world.

Furthermore, many large structures, such as exhibition halls, industrial plants, supermarkets, farms, etc., have a large footprint with few supporting pillars, resulting in a rooftop with low load-supporting capacity. Such rooftops require lightweight PV modules. If not, the cost of reinforcing such building before installing the heavy glass-based PV modules would make the renewable energy venture (building plus PV power infrastructure) inefficient and unappealing to potential business interests [5,9,10]. Lightweight PV has been extensively and comprehensively researched elsewhere as part of building-integrated PV and for predominantly urban building applications [5,6,7,8,11].

In addition, as silicon will be the predominant PV technology for the foreseeable future [5,6], we need to aid lightweight silicon-based PV modules. Structural strength, and impact resistance of lightweight PV modules, especially against heavy winds and hailstorms in four-season countries in Europe and North America [12,13], is one of the most important technological concerns in the development of lightweight PV modules. The front panels of a PV module can be made strong enough not to break under impact loads, such as hailstorms; nevertheless, the energy is transferred straight to the underlying material—first to the encapsulation (usually ethylene vinyl acetate), which would simply give way. The energy was then passed to the brittle silicon cells, which are particularly susceptible to such point impact loading, causing cracks to develop (nucleate) and/or propagate [7,13,14]. As a result, electrical performance degrades gradually or dramatically, which can lead to hotspots and potentially dangerous situations (such as fires, etc.).

Even though the idea of lightweight PV modules is tempting, nevertheless, it is not currently a practical option owing to concerns about structural stiffness and reliability [5,6,7,12,13]. Numerous commercial lightweight PV systems (even those that meet IEC/UL standards) have limited operational time [6,7]. Despite the fact that silicon cells remain internally delicate and extremely vulnerable to certain impact loads, recent studies with substantial material advances and ingenious design have allowed great improvements in the impact resistance of numerous polymer-based substrates used as front panels (as an alternative of glass) in conventional PV modules [13,14,15,16].

Natural structural materials, such as those found in mantis shrimp and nacre, have been demonstrated to provide superior mechanical and, particularly, impact resistant properties [17,18,19]. For example, the dactyl club of mantis shrimps has a 3D construction with a helicoidal shape that can disperse energy through quasi-plastic compression responses, providing a barrier to the spread of microcracks throughout recurring impacts [20,21,22,23]. Recent publications from our own research have stated higher impact resistance of such materials [24,25]. Further, the comparison in terms of mechanical properties of the 3D helicoidally aligned layered materials vs. layered materials without rotation offsets (or in other words, unidirectional layered materials) has been shown experimentally in our own previous publication by Agarwal et al. [26]. In addition, such comparisons have also been reported experimentally by other research groups using various materials (glass filament epoxy [27], carbon epoxy [27,28], and fiber sizes [27,28,29]). Agarwal et al. [26] reported for the smaller scale (using the custom electrospinning set up as an additive manufacturing methodology) of the other groups’ studies, in terms of fiber diameters. Furthermore, comparison in terms of mechanical properties of the 3D helicoidally aligned layered materials vs. a sample of same thickness as the layer stack (or in other words, bulk sample) has also been demonstrated experimentally in our own previous publication [26], in addition to other previous reports [27,28,29], chiefly from Kisailus et al. [29]. Again, our group’s studies—started by Agarwal et al. [26] to the more recent Sahay et al. [25] and Budiman et al. [30]—simply further pursued this line of investigation into the smaller scale fibers (using the custom electrospinning set up as an additive manufacturing methodology) and for the potential application of the unique materials for silicon-based PV module technology. The general outcome of such studies was that the layered structure of such materials, which consists of helicoidally aligned 3D fibers compared to typical layered structure/bulk material, would effectively absorb the impact energy and transfer very slight energy to the fragile silicon solar cells. This would allow novel lightweight PV module design with improved impact resistance and structural reliability (based on polymer materials for the front and back panel) particularly against cracks in the silicon cell.

The objective of this work is to provide a numerical analysis and modeling to predict how stress changes during impact loading in 3D-architectured layered polymer systems with helicoidally oriented fibers. This model demonstrates the fundamental feasibility of the proposed concept, namely, the use of 3D-architected layered polymer assemblies of helicoidally oriented fibers to protect silicon solar cells against the nucleation and proliferation of cracks caused by impact loads (e.g., from hailstorms) in the design of lightweight PV modules. We are expanding our approaches to enable this unique material for use in lightweight PV technologies. We are building on our earlier studies on novel materials [24,25,26,31] as well as numerical modeling of stresses in PV module design [32,33,34,35,36,37]. Furthermore, the design of lightweight PV modules would allow the integration of PV into curved or contoured surfaces, resulting in a more appealing design for integrating PV into urban structures.

## 2. Methodology

### 2.1. Material

The material used in making the multilayered composite plate is PVDF-HFP fibers, as was used in the experimental impact testing [30]. PVDF-HFP is polyvinyl alcohol (PVA) with MW = 98,000, polyvinylidene fluoride-co-hexafluropropylene (PVDF-HFP) with MW = 400,000, acetone, and dimethylacetamide (DMAc) were obtained from Merck, Singapore, as reported in our previous study [38]. More complete information about the materials used in the experimental impact testing can be found in [24]. 

The multilayered composite plate was modeled with fiber alignment in the layers changed from 90° (i.e., grid pattern) to 45° and then later to 15° to simulate the experiments (as illustrated in Figure 1), in which the impact resistance increase was observed [30]. Thus, in the present study, three multilayered composite plates were modeled:First composite plate (Composite Plate A) consists of layers with fiber alignments of [0°, 90°, 180°, 270°, 360°].Second composite plate (Composite Plate B) consists of layers with fiber alignments of [0°, 45°, 90°, 135°, 180°, 225°, 270°, 315°, 360°], as illustrated in Figure 1b.Third composite plate (Composite Plate C) consists of layers with fiber alignments of [0°, 15°, 30°, 45°, 60°, 75°, 90°, 105°, 120°, 135°, 150°, 165°, 180°, 195°, …, 360°]

The dimensions of the composite plate in the Finite Element Model (FEM) used are 100 mm × 100 mm × 0.4 mm (length × width × thickness). Impact loading uses a steel ball with a radius of 30 mm. The mass of the ball is 0.1 kg with an impact velocity of 1 mm/ms. These parameters scale with the experimental impact testing as reported in [30]. Mechanical properties of the PVDF-HFP fiber were obtained from our previous publication [24], which reported in detail the materials used in the experimental impact testing in Ref. [30] as well as in the present numerical simulation study (See Table 1).

### 2.2. Electrospinning-Based Additive Manufacturing (Es-AM)

Only recently has electrospinning-based additive manufacturing (Es-AM) technology enabled such intricate 3D designs [25,31]. Near-field electrospinning (NFES) has been used to fabricate helicoidally oriented fiber layers as an additive manufacturing approach [7,8]. Agarwal et al. [24] describe in detail the fabrication of helicoidally oriented fiber layers with different angular orientations. NFES typically creates helicoidally oriented fiber layers by depositing one-dimensional fibers at precise locations in a controlled manner and then stacking the fibers layer by layer with angular offsets to create a 3D helicoidally arranged synthetic structural composite (HA-SSC) (see Figure 1). Typically, in the case of 90° HA-SSC90, the fiber layers were deposited with 90° angular offsets starting from 0°, 90°, 180°, 270°, to 360°—as mimicked in this numerical simulation study as Composite Plate A. Similarly, in HA-SSC45 with 45° angular offsets, the fiber layers were deposited at 0°, 45°, 90°, 135°, 180°, 225°, 270°, 315°, 360°—as mimicked in this numerical simulation study with Composite Plate B, and schematically illustrated further in Figure 1b. In HA-SSC15 with 15° angular offsets, the fiber layers were deposited at 0°, 15°, 30°, 45°, 60°, 75°, 90°, 105°, 120°, 135°, 150°, 165°, 180°, 195°, …, 360°—as mimicked in the present study as Composite Plate C. The HA-SSCs samples were between 230 and 250 µm thick. In the experimental impact tests published in [30], we used only HA-SSC15 and HA-SSC45—with fiber orientation at rotation angles of 15° and 45°, respectively—from the different variants of the HA-SSC samples we reported in [24]. See [30] for more information on the impact testing of HA-SSCs.

This composite fabricated and reported in [24] is not yet ready for incorporation into PV power infrastructure. An optically transparent material is required for practical PV application with similar transmission of sunlight (in terms of intensity and range of suitable wavelengths). Nevertheless, as elucidated in the Introduction and the Materials section, the emphasis of the study is to establish the viability of the idea of improved impact resistance through the 3D-architected impact-resistant encapsulant, not its complete industrial incorporation into PV module design. 

### 2.3. Finite Element Modeling of the Impact Loading on Multilayered Composite with 3D Helicoidal Architecture

Finite element (FE) models allow us to estimate the mechanical stress that develops during impact loading of the multilayered fiber-based composite materials with the 3D helicoidal arrangement to predict the damage mechanisms associated with the stress evolution during impact loading. However, to keep computational complexities to a minimum while still gaining the fundamental deformation mechanics, we approximate the fiber-based composite layers with laminate geometry that has anisotropic mechanical properties in a certain direction in each layer. We used thin film geometry in the FE model [38]. We used commercially available general-purpose LS-DYNA (4.6.19, LSTC (Ansys, Inc.), Canonsburg, PA, USA) FE software to obtain the stresses induced in the multilaminate structures to understand the evolution of stresses during impact loading. The FE model uses shell elements of a conventional thin film. The thin film sample was modeled using regular 8-noded quadrilateral elements (CPEG8). A generalized plane strain condition was assumed. 

Figure 2 shows the multilayered composite plate, which in the LS-DYNA software was modeled using 4N plat on shape meshes. The size of the mesh is 2 mm. All the edges of plates are simply supported. The plates are subjected to impact loading on the middle of a surface in the form of a ball dropping onto the plate—mimicking the ball-dropping impact test as described in our previous report on a similar multilayered composite with 3D helicoidal architecture [30]. The impact loading was restricted to the z-direction. Data on displacement in the z-direction is collected on each node element of the middle layer of the multilayered composite plate. 

The mechanics of the thin film plate during the impact loading here were modeled with the surrogate approach [39]. The strain in the film here due to the impact loading was surrogated by the scaling approach (size of the ball with respect to size of the multilayered composite geometry are equivalent between the model and the experiment) to keep the complexity to a minimum, while still obtaining important insights about stress evolution and damage distribution mechanisms during the impact loading. Thus, the relative magnitude of the stresses with respect to time during impact loading represents the evolution of energy distribution during the ball-dropping impact testing, with actual silicon solar cells under the 3D-architected layered polymer structures consisting of helicoidally aligned fibers, as reported in [30]. Uniform deformation over the volume of each layer of the multilayered composite is assumed during the process—leaving only the asymmetric deformation due to the anisotropy of each of the laminate due to different rotational angle of fiber alignment in each layer of the 3D helicoidal architected polymer composite. Fully elastic behavior of each of the laminate was assumed, which is reasonable given the scaling approach and the actual experimental results [30]. Interfaces between laminates were modeled as surface-to-surface tie constraint [39].

## 3. Results and Discussion

### 3.1. Results of the Finite Element Simulation: Impact Contact between the Steel Ball and the Plate

The position of the steel ball (modeled as sphere) with respect to the composite plate for various time instances during the impact contact is shown in Figure 3 (from the side of the model for Composite Plate A). The steel ball impinges the plate at an initial velocity of 1 mm/ms. The simulation shows the sphere has started contact with the plate surface at the time of t = 1.4399 ms. The simulation then shows in Figure 3 that the steel ball has exerted sufficient force to deform the plate laterally at t = 1.8899 ms. Consequently, at time t = 5.5788 ms, the plate has reached its maximum deformation. Lastly, at the time instant of t = 9.9811 ms, the steel ball bounced back and lost contact with the plate. Although the above time evolution was shown for Composite Plate A, similar occurrences represent the impact contact evolution between the steel ball and the plate in the other composite plates studied in the present study (Composite Plates B and C).

### 3.2. Results of the Finite Element Simulation: Deformation of the Composite Plate

The distribution of the Max Principal Stress (or the first principal stress, S1) with respect to time during the impact loading is shown in Figure 4, Figure 5 and Figure 6 for Composite Plates A, B, and C, respectively. Initially, when the sphere first impinged on the plate, Figure 4, Figure 5 and Figure 6 all show that stress concentration at the plate center. This is expected, as it is where the impact of the sphere on the plate occurred. The stress that is initiated at the plate center spreads out quickly to the whole plate and is absorbed by the composite plate structures at different rates. 

Figure 4 clearly shows that the stress distribution during impact loading follows the four-symmetry that is created by the grid pattern of Composite Plate A, which consists of layers with fiber alignments of [0°, 90°, 180°, 270°, 360°]. Stress concentrated at the center of the plate and reaches the maximum principal stress (which occurred in all composite plates in the present study in the *z*-direction) of 0.095 GPa at t = 7.3798 ms. Over time, the stress reduces and finally reaches the zero-stress state after about 9.5 ms.

Figure 5 clearly shows that the stress distribution during impact loading follows the double four-fold symmetry that is created by the grid pattern of Composite Plate B, which consists of layers with fiber alignments of [0°, 45°, 90°, 135°, 180°, 225°, 270°, 315°, 360°], as illustrated in Figure 2b. Figure 5 also shows that the stress concentrated at the center of the plate and reaches the maximum principal stress (in the *z*-direction) of 0.035 GPa at t = 4.7384 ms. Over time, the stress reduces and finally reaches the zero-stress state after about 8 ms. These parameters clearly demonstrate a much higher dissipation rate of damage in Composite Plate B, as compared to Composite Plate A. The maximum principal stress shown here in Figure 5 (for Composite Plate B) is lower, and the rate in which the stress is distributed and subsequently reduced to zero upon the same impact loading (impact load and loading rate) is much higher. This suggests that Composite Plate B with 3D architecture consisting of layers with fiber alignments of [0°, 45°, 90°, 135°, 180°, 225°, 270°, 315°, 360°] is much more effective and efficient in absorbing and dissipating impact energy and damage compared to Composite Plate A. 

Figure 6 further suggests that Composite Plate C with 3D architecture consisting of layers with fiber alignments of [0°, 15°, 30°, 45°, 60°, 75°, 90°, 105°, 120°, 135°, 150°, 165°, 180°, 195°, …, 360°] is also much more effective and efficient in absorbing and dissipating impact energy and damage compared to Composite Plate A. It also exhibits an incremental increase in the absorption and dissipation rate of impact energy and damage compared to that of Composite Plate B. Figure 6 shows that the stress concentrated at the center of the plate and reached the maximum principal stress (in the *z*-direction) of 0.034 GPa at t = 4.6478 ms. Over time, the stress reduces and finally reaches the zero-stress state after about 7.5 ms. These parameters clearly suggest that the rotational angle of the 3D helicoidal architecture influences the impact resistance of the materials. The smaller rotational angles clearly play a key role in increasing the dissipation rate of the impact damage, although the increase could be moderated after some levels of rotational angles. This is further substantiated by the stress vs. time response data shown in Figure 7.

Figure 7 clearly indicates that the Composite Plates B and C have a much shorter time response in terms of absorbing the impact energy (and thus damage dissipation in the event of plastic deformation and fracture events). Furthermore, the much lower stress intensity in Figure 7b,c demonstrates that the smaller rotational angle steps are effective in quickly and efficiently distributing the stress concentration. Both Composite Plates B and C exhibit stress levels getting back to pre-impact level (zero stress state) within under 7.5 ms, compared to just under 10 ms for Composite Plate A. Composite Plates B and C also show maximal principal stress intensities of 0.035 GPa and 0.034 GPa, respectively. This is much lower stress level compared to that of Composite Plate A, which reaches 0.095 GPa (almost three times). It is evident through the stress evolution during the impact loading simulation that the smaller rotational angles play a key role in increasing the dissipation rate of the impact damage, although the increase could be moderated after some levels of rotational angles. The increase in the absorption rate of impact damage and dissipation rate of stress intensity with reduction of rotational angle in the 3D helicoidal architecture seems to taper off after 45°. The FE simulation was conducted with the surrogate approach, thus while the absolute magnitudes still need to be further verified with experimental study using the same scales as in the modeling, the relative comparison between the effects of rotational angles in increasing the rate of absorption of damage and dissipation of stress intensity suggests self-consistency and excellent agreement with the experimental impact loading as reported in our earlier publication [30] and as will be further elaborated in the following section.

### 3.3. Comparison with Impact Test of Photovoltaic (PV) Cells

The experimental study on the impact-loading test of the 3D helicoidally architected polymer composite materials (with different rotational angles) has been reported in our recent publication [30]. In it, we use fragile silicon solar cells, which are highly susceptible to impact load underneath the polymer composite materials, such that the polymer composite materials act as a protection layer for the fragile silicon solar cells [30]. A customized impact testing setup (as illustrated in Figure 2b) was used to determine the impact resistance of such solar cells when protected by the samples [40]. More complete information about the impact test using the steel ball dropping method can be obtained in [30].

The fracture height indicates the height at which we began observing fracture of the silicon solar cells under the polymer composite materials. The nominal height of the bare Si solar cell group was found to be 25 (±5) cm, which was the real height at which all silicon cells fracture in the at least six times we repeated the tests (we performed the ball drop tests to more than six samples at this height—up to 11 samples). The uncertainty level of the experiment was ±5 cm. Consequently, the fracture data indicated the heights at which the fracture started. Heights greater than the abovementioned value would evidently fracture the silicon cells.

The fracture heights when we had typical encapsulant: ethylene vinyl acetate, on top of the silicon solar cells, was 50 ± 4 cm. The fracture heights are 69 ± 2 cm and 82 ± 4 cm for the Composite Plate B: HA-SSC15 and Composite Plate C: HA-SSC45, respectively [24,25]. Therefore, it was evident that the HA-SSC materials were the better protector of the silicon solar cells against impact loading, compared to the nominal EVA. Both HA-SSC composites (HA-SSC15 and HA-SSC45, respectively, Composite Plates B and C) enable significantly higher fracture heights and related specific potential energies (well beyond experimental uncertainties) before the underlying silicon solar cells fracture or trigger/spread catastrophic fracture events in ball drop tests. Because of their helicoidally oriented fiber-reinforced layered structures, both HA-SSC would efficiently disperse the impact energy and deflect the crack laterally by following the helicoidal progression of fiber directions, rather than immediately fracturing in a straight line through the thickness of the HA-SSCs [24,25,41]. Therefore, a smaller amount of impact energy is transferred to the underlying silicon solar cell.

The fractography images described in Ref [30] show abovementioned efficient mechanism for load dissipation as well as effective damage/energy absorption. In addition, both HA-SSC permit multiple crack lines on the silicon cells, signifying high dissipated impact energy and load transfer to the side (as evidenced by crack lines at different angles on the silicon cell surfaces). In typical monocrystalline silicon wafers, cracks seem to track the favored crystallographic directions of <110> linked to the weakest crystallographic planes of {111}, as has been extensively stated in the literature for both PV and other silicon-based devices [32,33,34,35,36,37,42,43].

The crack regularly encounters variations in the modulus of the fibers and the matrix material. The crack in HA-SSC travels along one fibrous layer, encounters a modulus deviation due to the matrix material present, penetrates further into the matrix, then reaches another fibrous layer in a dissimilar direction and plane, and finally diverges from its original route to follow a dissimilar path. The crack proliferates in multiple planes and orientations, rotates and twists inside and outside the fiber and matrix phases, and places greater stress on the fibers short of catastrophic failure. Due to the helicoidal network of the fibers in the composites, more energy is required for the propagation of the crack, so only a limited amount of impact energy/damage is transferred to the underlying silicon solar cells. According to Budiman et al. [30], the Composite Plate B: HA-SSC45 appears to be the most successful at consistently deflecting fracture/ impact damage along different angular orientations. The fracture heights show that the solar cells fracture at 82 ± 4 cm, which is the highest value among the samples investigated in this study. However, the study [35] paints a somewhat different depiction for the Composite Plate C: HA-SSC15. The breakage of the solar cells under HA-SSC15 happens at a lower height (69 ± 2 cm), resulting in a lower specific potential energy (and thus a lower impact energy/damage absorption rate). 

In the present study, our FE simulation shows that the absorption rate of impact damage and dissipation rate of stress intensity increases with reduction of rotational angle in the 3D helicoidal architecture, although the effect seems to taper off after 45°. The FE simulation here shows that Composite Plate C: HA-SSC15 exhibits faster absorption rate of impact damage and stress level reduction (back to pre-impact state), although by a much smaller margin compared to that between Composites Plate A and B (from rotational angle of 90° to 45°). While this may not be in general agreement with our previous study reported in Budiman et al. [30], the FE simulation findings in the present study do agree very well with another earlier experimental report from our group, as described in Agarwal et al. [24]. The difference between these two studies is silicon solar cells were used under the HA-SSC in the impact loading experiments in Budiman et al. [30], while glass coverslips were used in Agarwal et al. [24]. The HA-SSCs used were, however, the same [30].

We believe that the variance (in the ability to absorb impact damage between HA-SSC15 and HA-SSC45) is due to the monocrystalline silicon solar cells that were among the samples of HA-SSC, because we followed the same method of the ball-dropping test as [24,25], together with the identical size of the steel ball. Compared to the glass slip, the monocrystalline silicon samples exhibit a crystallographic dependency of the mechanical properties, such as the preferential occurrence of fractures. Consequently, the whole relationship amongst the helicoidal orientation in HA-SSC and fracture of monocrystalline silicon solar cells needs to be further explored, which is outside the scope of the current FE simulation. Nevertheless, FE simulation in the present study takes into consideration the fact that the material put under the HA-SSC was a silicon monocrystalline solar cell (with its anisotropy in the mechanical properties), and not a glass coverslip (which has mostly isotropic mechanical properties).

### 3.4. Enabling Next-Gen Lightweight Photovoltaic (PV) Module Technology

It is clear from the experimental results [24,30] and the FE simulation shown in the present study that the HA-SSCs are excellent at absorbing and dispersing impact energy/damage, protecting the delicate solar cells underneath from point impact loads to which silicon PV is particularly susceptible. The objective of this paper is to complement the previously published experimental data with FE simulation study on the basic viability of using HA-SSCs for PV encapsulation to shield the sensitive solar cells, particularly in the design of lightweight PV modules that are predominantly susceptible to impact loadings, such as hailstorms, as outlined in IEC 61215/61646 clause 10.17. Although many current studies have revealed that a variety of polymer-based materials [12,13,14,15,16] can be used to improve the fracture and impact resistance of lightweight PV modules; nevertheless, the encapsulation materials employed in these studies were all EVA—albeit with different thicknesses or slight variations, such as low curing temperature [44].

Due to the economic impact and production maturity of the entire lightweight PV module, EVA was clearly selected as the encapsulation material of choice in previous studies. Our previous experimental results, as well as the numerical simulation results reported in this manuscript, provide initial indication from the standpoint of basic technological viability for the use of other types of novel polymer films (i.e., not EVA) with unique 3D architectures to allow strong and impact resilient lightweight PV module designs.

Nevertheless, there are many potential technological challenges to be resolved that prevent this unique idea from being fully realized. First, the HA-SSCs created were not transparent. Transparent protective layers on silicon cells in PV modules are an absolute must. It should be emphasized that the tests [30] and modeling of FE in the current study were performed to confirm the feasibility of integrating HA-SSC into PV modules. After our FE simulation results demonstrated the basic, complete feasibility using the electrospinning-based additive manufacturing (AM) method, we were able to identify additional polymers that we could fabricate in transparent systems, such as nylon [45] and poly(methyl methacrylate) (PMMA), or PMMA-based composites [46,47]. Furthermore, not only the interfacial adhesion with the front panel but also with the solar cell itself must be adequate [44]. The interfacial adhesion was not modeled in the current work and is the next step in the study. To maintain the 3D design throughout the lamination process, these novel materials may need further development [44,48,49]. All these may pave the way for future research of innovative polymeric composites/materials with 3D architecture to improve the use of lightweight PV modules and technologies.

## 4. Conclusions and Future Perspectives

In this FE simulation study, fiber-based 3D composites with helicoidally architecture were revealed to have outstanding impact energy/damage absorption and high-energy dissipation rate. Therefore, when applied to solar cells, they offer better shield against impact loads. Composite Plates C and B permit considerably higher fracture heights of 69 ± 2 and 82 ± 4, respectively, in comparison to 25 ± 5 for unprotected solar cells, and 50 ± 4 for EVA-protected solar cells, as observed during the ball-drop experiment. These helicoidally aligned synthetic composites (HA-SSCs) were fabricated using an electrospinning-based additive manufacturing (AM) technology that has only recently become possible at our group (Xtreme Materials Laboratory) in the past few years. During the ball-drop impact simulation using the Finite Element (FE) method, the HA-SSC composite materials (Composite Plates A, B, and C) showed consistent increase in damage absorption rate and stress level dissipation mechanism with reduced rotational angle of the HA-SSC materials. This is in excellent agreement with reports in the literature using isotropic materials (such as amorphous glass), although the agreement may be limited for anisotropic materials (such as monocrystalline silicon). The current FE simulation results add to the growing body of data that the innovative HA-SSC could be used for PV encapsulation to allow the design and fabrication of lightweight PV modules. The FE simulation results, as reported in the present manuscript, indicate impact protection increased with the reduction in the azimuthal angle to 45°, after which the impact protection remained more or less constant. This is a unique insight which could have important implications for the silicon-based PV technology industry, as well as for other societally important applications of the HA-SSC materials, such as for lighter army combat vests and sports gear (helmets, etc.).

## Figures and Tables

**Figure 1 polymers-14-01228-f001:**
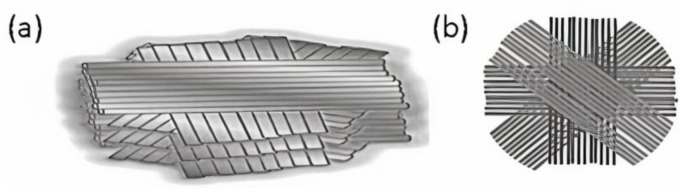
Schematic illustration of the multilayered composite (**a**) consisting of layers with 3D-architected helicoidally aligned fibers; (**b**) with fiber alignments of [0°, 45°, 90°, 135°, 180°, 225°, 270°, 315°, 360°] described in our previous report [24].

**Figure 2 polymers-14-01228-f002:**
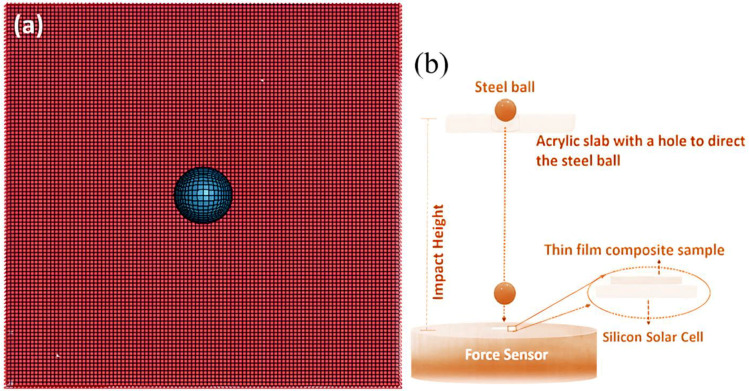
Schematic of the 2D plane–strain FE model (**a**) with thin film shell elements of multilaminate structures under impact loading mimicking the ball-dropping impact test (**b**) as described in our previous report [30].

**Figure 3 polymers-14-01228-f003:**
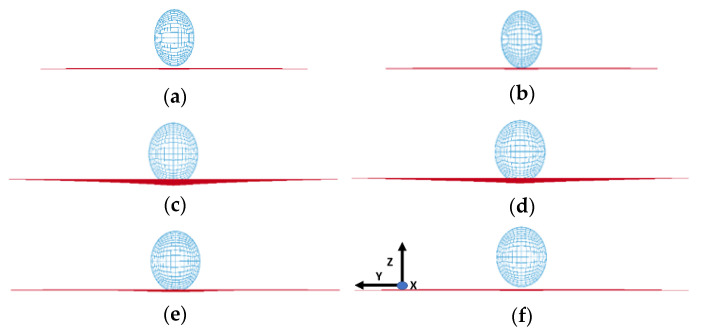
FE simulation showing impact contact between the sphere and the composite plate, (**a**) t = 0.4388 ms, (**b**) t = 1.4399 ms, (**c**) t = 1.8899 ms, (**d**) t = 5.5788 ms, (**e**) t = 7.8338 ms and (**f**) t = 9.9811 ms.

**Figure 4 polymers-14-01228-f004:**
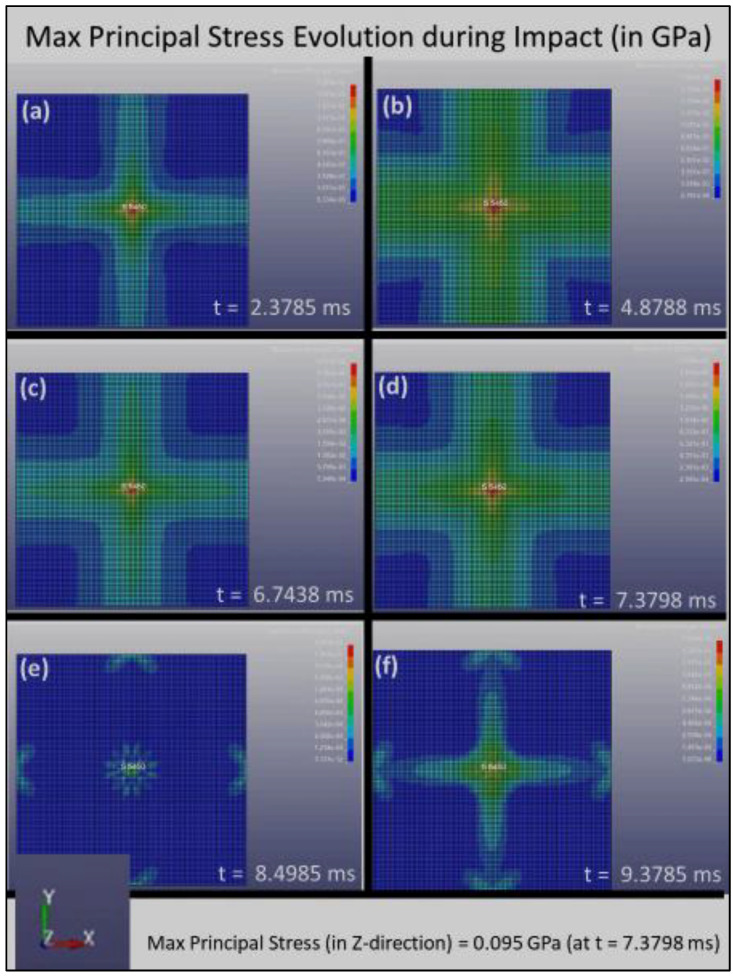
The distribution of the maximum principal stress, S1, with respect to time for Composite Plate A during impact loading, (**a**) 2.3785 ms, (**b**) 4.8788 ms, (**c**) 6.7438 ms, (**d**) 7.3798 ms, (**e**) 8.4985 ms and (**f**) 9.3785 ms.

**Figure 5 polymers-14-01228-f005:**
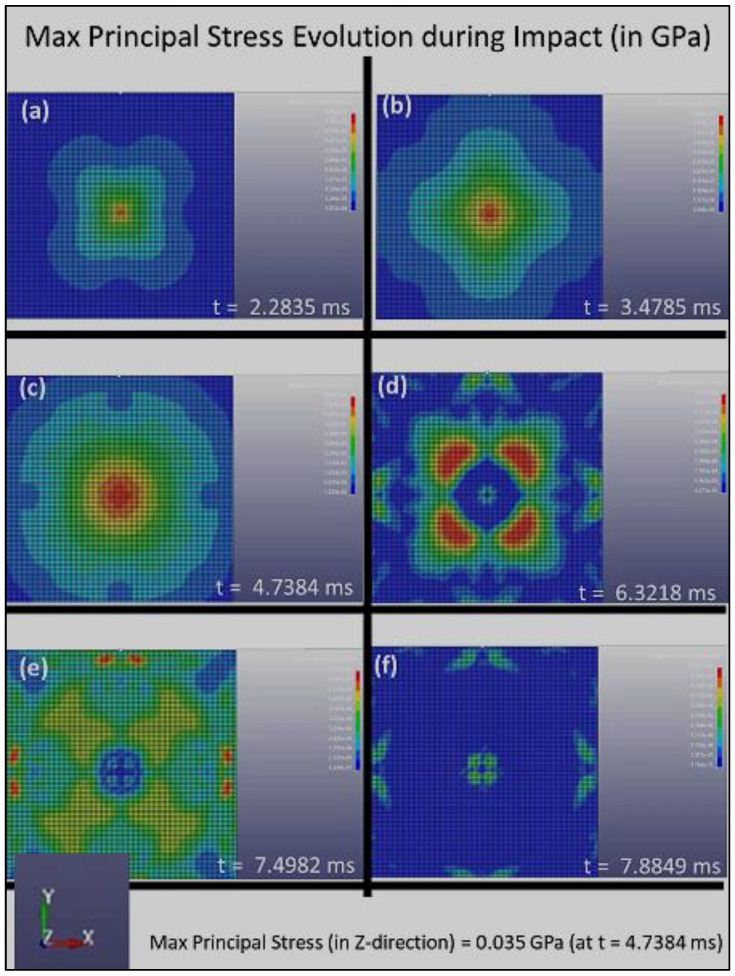
The distribution of the maximum principal stress, S1, with respect to time for Composite Plate B during impact loading, (**a**) 2.2835 ms, (**b**) 3.4785 ms, (**c**) 4.7384 ms, (**d**) 6.3218 ms, (**e**) 7.4982 ms and (**f**) 7.8849 ms.

**Figure 6 polymers-14-01228-f006:**
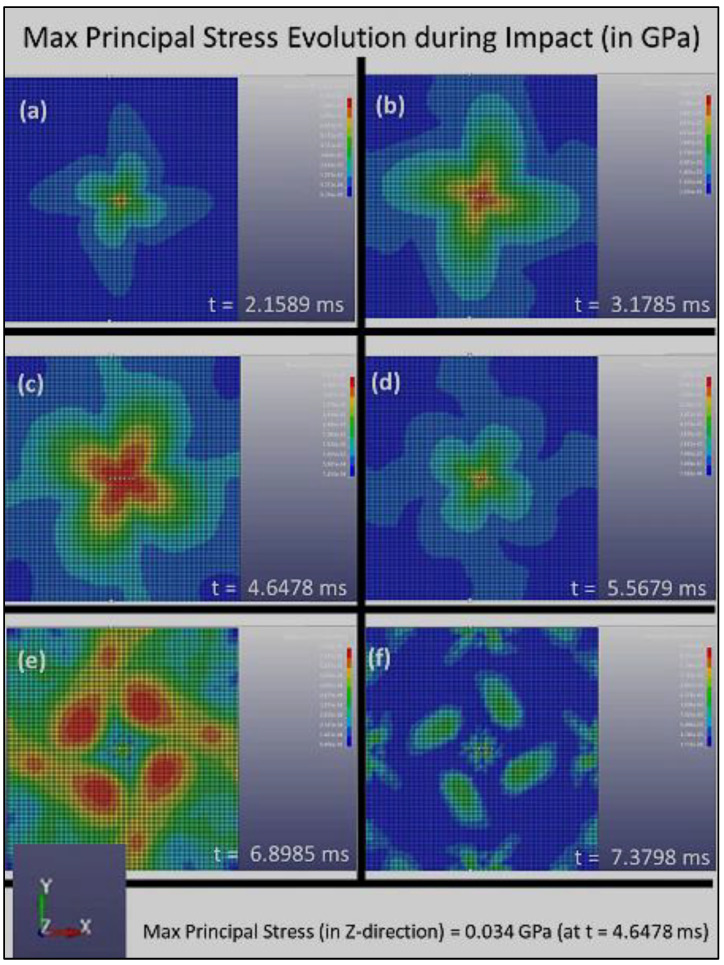
The distribution of the maximum principal stress, S1, with respect to time for Composite Plate C during impact loading, (**a**) 2.1589 ms, (**b**) 3.1785 ms, (**c**) 4.6478 ms (**d**) 5.5679 ms, (**e**) 6.8985 ms and (**f**) 7.3798 ms.

**Figure 7 polymers-14-01228-f007:**
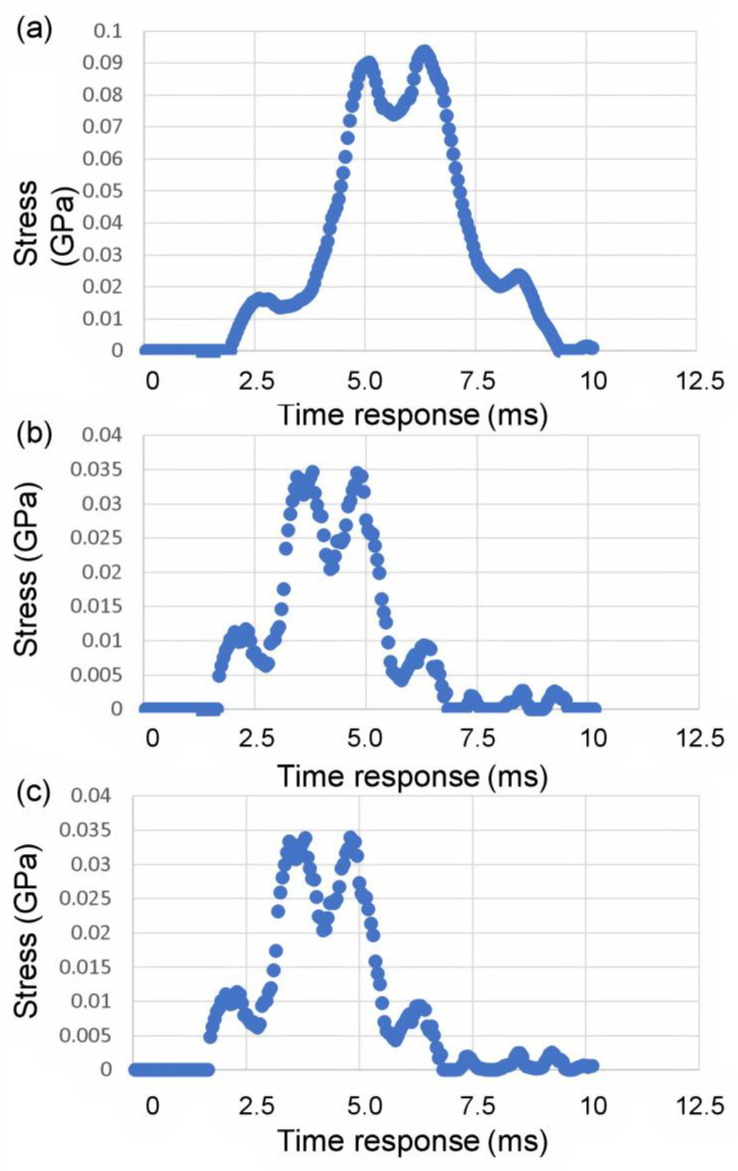
The time response of the distribution of maximum principal stress, S1, for (**a**) Composite Plate A, (**b**) Composite Plate B, and (**c**) Composite Plate C, during impact loading.

**Table 1 polymers-14-01228-t001:** Mechanical properties of PVDF-HFP fiber composite material (in each layer, with 0° and 90° represent longitudinal and transverse direction, respectively, of the fibers) used in simulation modeling [24].

Properties	Simbol	Unit	PVDF-HFP Fiber
Young modulus 0°	E1	MPa	70
Young modulus 90°	E2		30
Poisson ratio	V12		0.1
Ultimate tensile strength 0°	Xt	MPa	60
Ultimate compression strength 0°	Xc		57
Ultimate tensile strength 90°	Yt	MPa	30
Ultimate compression strength 90°	Yc		27
Ultimate tensile strain 0°	ext	%	85
Ultimate compression strain 0°	exc		80
Ultimate tensile strain 90°	eyt	%	45
Ultimate compression strain 90°	eyc		35
Density	ρ	g/cm^3^	1.6

## Data Availability

The data presented in this study are available upon request from the corresponding author.

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
