# Peer review of "Modeling Impact Mechanics of 3D Helicoidally Architected Polymer Composites Enabled by Additive Manufacturing for Lightweight Silicon Photovoltaics Technology"

_polymers, 2022, doi:10.3390/polym14061228_

Round 1

Reviewer 1 Report

In the  manuscript entitled Modeling Impact Mechanics of 3D Helicoidally Architected Polymer Composites Enabled by Additive Manufacturing for Lightweight Silicon Photovoltaics Technology is focused on the  modelling of results previously published by same authors.

The authors pointed out the attention of the 3D helicoidal design for impact protection of the PV module. In the discussion is difficult to find the rationale of the design and the helicoidal fabricated structure.The simulations seem not new and most of the data can be found in the literature (some of them published by same authors).

I found the manuscript with a poor level of novelty. Literature reports different similar works. Some of them are published by same authors

References section seems to be prepared of another journal style.

Some of the Figures can be found in other published papers.

Author Response

A file containing answers to the reviewer's questions attached.

Reviewer 2 Report

Greetings, Editor thank you for providing me with the opportunity to review the article. I reviewed the article with title Modeling Impact Mechanics of 3D Helicoidally Architected Polymer Composites Enabled by Additive Manufacturing for  Lightweight Silicon Photovoltaics Technology. The article topic is intriguing and promising in the area. Overall, the article structure and content are suitable for the Polymers journal. I am pleased to send you major level comments. Please consider these suggestions as listed below.

  1. The title is too long please concise and revised it. but the abstract seems to be fine. Please add one more introductory line of your objective in beginning of abstract.
  2. Research gap should be delivered on more clear way with directed necessity for the future research work.
  3. Introduction section must be written on more quality way, i.e., more up-to-date references addressed. Please target the specific gap such as 2015-2021 etc.
  4. Why some paragraphs are in red colour, and some are green, black. What is the difference?
  5. The novelty of the work must be clearly addressed and discussed, compare previous research with existing research findings and highlight novelty.  
  6. What is the main challenge? Why author choose this material? Please highlight in the introduction part.
  7. Please include all chemical/instrumentation brand name and other important specification. Please follow the Polymers manuscript preparation guidelines. The arrangement of methodology and material is wrong.
  8. The main objective of the work must be written on the more clear and more concise way at the end of introduction section.
  9. Please check the abbreviations of words throughout the article. All should be consistent.
  10. Please provide space between number and units. Please revise your paper accordingly since some issue occurs on several spots in the paper. Please also write the units in scientific way such as cm3.
  11. Please add chemical reagents section and stated all chemical with brand specifications.
  12. Regarding the replications, authors confirmed that replications of experiment were carried out. However, these results are not shown in the manuscript, how many replicated were carried out by experiment? Results seem to be related to a unique experiment. Please, clarify whether the results of this document are from a single experiment or from an average resulting from replications. If replicated were carried out, the use of average data is required as well as the standard deviation in the results and figures shown throughout the manuscript. In case of showing only one replicate explain why only one is shown and include the standard deviations.
  13. Please provide high quality figure 1 and 3.
  14. Please add a comparative profile section to compare your results and prove how it better than previous.
  15. Section 4 should be renamed by Conclusion and Future perspectives. Conclusion section is missing some perspective related to the future research work, quantify main research findings, highlight relevance of the work with respect to the field aspect.
  16. To avoid grammar and linguistic mistakes, minor level English language should be thoroughly checked. 
  17. Reference formatting need carefully revision. All must be consistent in one formate. Please follow the journal guidelines.

Author Response

A file containing answers to reviewer's question attached

Reviewer 3 Report

1. The authors could carefully double-check the format in the references which could be unified. 2. Some errors were highlighted in fluorescent as attached manuscript. The authors could double-check the correctness before re-submission.

Author Response

(The authors gave the same response as above.)

Reviewer 4 Report

The results from previous studies of the collective provided the basic data for the choice of model conditions, which are referred to several times in the present work. In this way, it can be concluded that the model is consistent with the results from previous studies and is therefore appropriate. The finite element method also allows a good study of the load behaviour over time, showing very interesting stress curves and stress maxima for the three composite plates with different layouts. The evaluation of the results obtained is fair and the conclusions are correct. It is correctly concluded that the overall relationship between the helicoidal orientation in HA-SSC and the fracture of monocrystalline silicon solar cells needs to be further investigated. It is a unique finding that the impact resistance increased with decreasing azimuthal angle up to 45°, after which the impact resistance remained more or less constant. The feasibility of integrating HA-SSC into PV modules has been demonstrated and points to future challenges in this area.

Line 91-98, and line 98-107 paragraphs contain duplications, please revise.

Since the figures should be self-explanatory, I would consider it fortunate if the colour symbol explanations in Figures 4-6 were given in one place, legibly.

Author Response

(The authors gave the same response as above.)

Round 2

Reviewer 2 Report

In the revised manuscript authors carefully considered all of the raised comments and the manuscript is much improved. It can be accepted in the present form.

This manuscript is a resubmission of an earlier submission. The following is a list of the peer review reports and author responses from that submission.

Round 1

Reviewer 1 Report

In this manuscript, the authors have presented 3D helicoidally architected fiber-based polymer composites in a FE simulation study to lead to excellent impact energy/damage absorption and dissipation rate. Various results have been presented, e.g. Impact Contact between the Sphere and the Plate, Deformation of the Composite Plate, Comparison with Impact Test of Photovoltaic (PV) cells. The manuscript was well written and structured, and the results were compared with literature for a promising result.

However, the difference and the reliability of the results from the FE model and the experiments should be more discussed, instead of “as reported in our previous study”, because the main results in the current manuscript was obtained just using the Finite Element simulation.

Reviewer 2 Report

 I found the manuscript with a poor level of novely. Literature reports different similar works. Some of them are published by same authors. So, i better description of the differences should be done in order to justify a new publication.

Reviewer 3 Report

The present manuscript „Modeling Impact Mechanics of 3D Helicoidally Architected Polymer Composites Enabled by Additive Manufacturing for Lightweight Silicon Photovoltaics Technology” by Budiman et al. submitted to mdpi-polymers is a follow up study of the already published paper in mdpi-polymers “Impact-Resistant and Tough 3D Helicoidally Architected Polymer Composites Enabling Next-Generation Lightweight Silicon Photovoltaics Module Design and Technology”.

The present manuscript is about modelling the previously obtained and published experimental results, reference 35 in the present manuscript. Due to this fact, images have been reused: Fig. 3 present is the same as Fig. 1 in the previous paper, Fig. 1b here is the same as Fig. 2 in the previous paper. The authors should make suitable referencing / copyright transfer – otherwise this is self-plagiarism.

The authors emphasize the 3D helicoidal architecture of the polymer composite used for impact protection of the silicon solar cells. It remains unclear, how this helicoidal architecture is considered within the modelling. Indeed, 3 different composite plates named A, B and C are considered. These composite plates consist of several layers which have been deposited for different azimuthal rotational positions of the substrates. Within each plate, the electrospun fibers are described to have a certain azimuthal angular alignment. But where is the outlined helicoidal architecture?

To my understanding, the relative azimuthal orientation of aligned fiber layer plays a role for the stress distribution and potential impact load protection. To work out if this layering with different azimuthal rotation actually is beneficial, the authors should present data of (1.) a layer stack without rotational offsets and (2.) a slab of same thickness as the layer stack.

Furthermore, a quantitative comparison with conventional impact load protecting materials based on the simulation method should be made. Indeed, the authors briefly mention EVA as standard protection material and give a value for “fracture height” (lines 350 and following) that is lower than for Composite Plate B and C. What is the value for Composite Plate A? How can this “fracture height” be extracted from simulation of stress distribution?

All over, the simulations of the present manuscript to not provide new insight into the topic compared with the already published experimental data. New insight could be that all possible azimuthal layering angles are simulated to make the best choice for impact protection, i.e. show that simulations help to reduce trial-and-error lab experiments.

Based on the shortage of new insight, I consider the interest to the readers in the present manuscript as minor, and therefore, I do not suggest the manuscript for publication.